# Immunotherapy for Soft Tissue Sarcomas: Anti-PD1/PDL1 and Beyond

**DOI:** 10.3390/cancers15061643

**Published:** 2023-03-07

**Authors:** Mina Fazel, Armelle Dufresne, Hélène Vanacker, Waisse Waissi, Jean-Yves Blay, Mehdi Brahmi

**Affiliations:** 1Centre Léon Bérard, 28 Rue Laënnec, 69008 Lyon, France; 2Faculté de Médecine Lyon Est, Université Claude Bernard Lyon, 8 Avenue Rockefeller, 69008 Lyon, France

**Keywords:** soft tissue sarcomas, immunotherapy, immune checkpoint blockade, adoptive T-cell therapies

## Abstract

**Simple Summary:**

Although immunotherapy has revolutionized the standard of care of many cancers, its efficacy in soft tissue sarcomas has been disappointing so far. Nevertheless, some recent studies have reported meaningful activity in a few selected histotypes, especially alveolar soft part sarcoma (ASPS). Furthermore, emerging biomarkers, such as the presence of tertiary lymphoid structures, seem to be predictive of the efficacy of immune checkpoint inhibitors. Finally, innovative therapeutic agents (especially adoptive T-cell therapies) and the combination of immunotherapeutic agents with other therapies such as tyrosine kinase inhibitors represent promising prospects.

**Abstract:**

Sarcomas gather a heterogeneous group of mesenchymal malignant tumors including more than 150 different subtypes. Most of them represent aggressive tumors with poor prognosis at the advanced stage, despite the better molecular characterization of these tumors and the development of molecular-driven therapeutic strategies. During the last decade, immunotherapy has been developed to treat advanced cancers, mainly thanks to immune checkpoint inhibitors (ICI) such as anti-PD1/PDL1 and later to adoptive immune cell therapies. In this review, we aim to summarize the state of the art of immunotherapy in soft tissue sarcomas (STS). Overall, the clinical trials of ICI that included a wide diversity of STS subtypes reported limited efficacy with some outlying responders. Both emerging biomarkers are of interest in selecting good candidates and in the development of combination therapies. Finally, the recent breakthroughs of innovative adoptive therapies in STS seem highly promising.

## 1. Introduction

Sarcomas gather a wide and heterogeneous group of rare malignant tumors of mesenchymal origins, representing less than 1% of all adult malignant tumors and 20% of childhood solid cancers [1,2,3]. They may occur on any site, at any age and are usually divided into soft tissue sarcomas (STS) and bone sarcomas. Furthermore, significant heterogeneity remains within these two subgroups, so that more than 150 distinct subtypes are described in the latest (fifth) WHO (World Health Organization) Classification of Soft Tissue and Bone Tumors [4]. Complete en-bloc R0 surgery is the cornerstone of the management of localized STS whereas the treatment of advanced STS remains challenging due to the rarity and the clinical and biological heterogeneity of these diseases [5]. Patients are treated with conventional chemotherapy (doxorubin, ifosfamide, etc.) or antiangiogenic tyrosine kinase inhibitors (TKI) (pazopanib), with a 3–6-month median progression free survival (PFS) and an 18-month median overall survival (OS) [6,7,8]. Therefore, advanced/metastatic STS represent a high unmet medical need.

Modern immunotherapy has been flourishing in the last decade, witnessing a significant expansion in the treatment of many solid and hematologic cancers [9]. Sarcomas were first suggested as good candidates, due to an historical rationale. Indeed, the first immunotherapy was practiced in the 19th century by Coley [10], with the inoculation of erysipela samples directly into a sarcoma during surgery. He then observed tumor shrinkage, which is believed to be induced by the recruitment of immune cells. Nevertheless, early experiences with immunotherapy in trials recruiting unselected STS subtypes have been disappointing. Regardless, significant efficacy has been observed in a subset of patients and/or with genetically modified T cell-based adoptive immunotherapy approaches.

We conducted a literature review to describe the current approaches to immunotherapy in STS. Here, we discuss the current state of immunotherapy for STS, the ongoing clinical trials evaluating immune checkpoint inhibitors (ICI), alone or in combination, or adoptive cellular therapy and the investigations to identify predictive biomarkers.

## 2. Immune Checkpoint Inhibitors in STS: Therapeutic Options

### 2.1. ICI Trials (Mono or Dual Blockade)

During the last five years, a few clinical trials have been conducted to evaluate ICI in STS, targeting regulators of the immune response by blocking the PD1/PDL1 and CTLA4 axes. In the last decade, ICI have dramatically changed the vision of cancer management by emphasizing the importance of the immune system in cancer growth, contrasting with the previous, inefficient immunotherapy mostly based on the use of vaccines. Each step in T cell-mediated immunity is regulated by counterbalancing inhibitory signals, which are commonly overexpressed in tumor cells or in the tumoral microenvironment. Immune checkpoint receptors expressed by T cells or antigen-presenting cells, such as PD1/PDL1 and CTLA4, when combined to their ligand, induce immune tolerance. ICI are antibodies that target and prevent the inhibitory ligand–receptor interaction and can therefore activate the immune system [11] (Figure 1).

The results of these studies remain modest overall, in light of the fact that they include all patients with no preselection. The first study conducted by the MSKCC (Memorial Sloan Kettering Cancer Center) evaluated ipilimumab (anti-CTLA4) in patients with metastatic synovial sarcomas [12]. This study was closed prematurely due to the lack of both inclusion and efficacy, with only six patients enrolled and no objective response (OR). The PEMBROSARC trial, conducted by the French Sarcoma Group (FSG), was therefore the first completed phase II study to assess the efficacy of pembrolizumab (anti-PD1) combined with metronomic cyclophosphamide, to convey its immunomodulatory properties [13]. The results were disappointing, with a partial response (PR) of only 1/50. Despite the negative result, this study has fueled the first hypotheses concerning the mechanisms of immune evasion, in particular via macrophage infiltration and the activation of the IDO (indoleamine 2,3 dioxygenase) pathway. Shortly after, the American phase II study SARC028 tested pembrolizumab as a monotherapy on 80 patients, half of whom had STS and the other half of whom had bone sarcomas [14]. Seven out of 40 patients (18%) with STS had an OR, including 4/10 patients with undifferentiated pleomorphic sarcoma (UPS) and 2/10 patients with dedifferentiated liposarcoma (DD-LPS). Finally, the third pivotal phase II study was Alliance A091401, evaluating nivolumab (anti-PDL1) alone or with ipilimumab [15]. An OR was achieved in 5% (N = 2/38) of patients in the nivolumab group and in 16% (N = 6/38) of patients in the combination group, including patients with ASPS, angiosarcoma, UPS and DD-LPS. In addition, expansion cohorts have been developed to explore the activity of UPS and DD-LPS with response rates of around 23% and 10%, respectively [16]. Overall, these results suggest that immunotherapy might have a therapeutical impact on STS, especially in combination with other therapies, which highlights the need for the identification of a predictive biomarker.

Furthermore, immunotherapy seemed to cause meaningful clinical activity in some selected histological subtypes, especially ASPS, an ultra-rare subtype of STS [17,18]. A neoepitope arising from the ASPL::TFE3 fusion protein itself has been speculated to be immunogenic [19,20], even though the exact rationale for their increased sensitivity to immunotherapy remains unclear. As of today, more than 150 patients with ASPS have been treated in clinical trials evaluating anti-PD1/PDL1, with response rates ranging from 7.1% to 54.5% depending on the study [21,22]. The more recent study was a single-arm phase II study that evaluated atezolizumab in ASPS patients, showing an OR rate of 37.2% (16/43). Importantly, the median duration of the confirmed response was 16.5 months (range: 4.9–38.1 months). Although there is no clear predictive factor for treatment response, some hypotheses arise, such as the prior failure of TKI [23] or CD8+ infiltration and PD-L1 expression [21], but investigations need to be pursued, possibly through genomic and transcriptomic landscape evaluation. In light of these results, the Food and Drug Administration (FDA) recently approved atezolizumab for patients with unresectable or metastatic ASPS (9 December 2022). As for angiosarcomas, multiple case reports find a meaningful activity signal [24,25,26,27], which is partly explained by a mutational profile linked to DNA damage caused by UV exposure, as is the case of face and scalp cutaneous angiosarcomas [28]. An angiosarcoma cohort was extracted from the trial evaluating dual inhibition by anti-CTLA4 and anti-PD1 in rare tumors (DART) [29]. The ORR observed was 25% among the 16 studied patients. The 6-month PFS rate was 38%. A confirmed response was observed in 60% of patients (3/5) with primary cutaneous scalp or face tumors, supporting the theory about UV exposure. Otherwise, in a phase II trial, single-agent pembrolizumab caused significant and prolonged antitumor activity in a few selected rare sarcoma histotypes, mostly in ASPS (8 of 17 patients with an objective response), but also in chordoma and SMRT (SMARCA4 deleted/malignant rhabdoid tumor) [30].

All those data support ICI (mono or dual blockade) as efficient therapeutics only in patients with selected subtypes of STS. Nevertheless, we are still facing the limited efficacy of classical ICI in most STS trials, and those disappointing results are largely related to non-optimal patient selection. Furthermore, other strategies consist of combining ICI with other classical treatments.

### 2.2. Combination with Tyrosine Kinase Inhibitors

Several clinical trials have assessed the association of ICI with TKI, in particular targeting the VEGF pathway. In a phase II study combining the pan-VEGFR inhibitor axitinib with pembrolizumab, some remarkable responses or ORR were observed in 7 of 12 patients with ASPS (58%), but only two responses out of 21 (10%) among other STS patients (1 with epithelioid sarcoma and 1 with leiomyosarcoma) were observed [22]. Importantly, all patients with ASPS who had biopsy samples in this study expressed PD-L1 and most had high tumor lymphocyte infiltration scores, consistent with the so-called “inflamed phenotype” observed in melanoma and other cancers that respond to ICI. The median PFS was 12.4 months (95% CI 2.7–22.3) in ASPS patients, and the combination showed acceptable toxic effects, consistent with previous clinical trials of the drugs used as monotherapies. Another phase II study assessing the combination of sunitinib with nivolumab included 58 evaluable patients with STS. The objective response rate was 21%, mainly including patients with ASPS and angiosarcomas but also clear cell sarcomas, extraskeletal myxoid chondrosarcomas and synovial sarcomas. The PFS at 6 months was 48% and the 18-month overall survival proportion was 100% for those with an objective response [31]. Many additional studies are underway. Nevertheless, with the response rate remaining low in an unselected population, it remains crucial to identify biomarkers to help in therapeutic decisions. Furthermore, a trial comparing ICI alone versus ICI plus antiangiogenic therapy would be of interest.

### 2.3. Combination with Chemotherapy

Chemotherapy has been proposed to overcome resistance to immunotherapy in sarcomas, aiming at rewiring the immune-cold/immunosuppressive microenvironment. Clinical trials have tested agents that promote T-cells while downregulating Treg lymphocytes, immunosuppressive tumor-associated macrophages or immunosuppressive cytokines, thanks to a direct immune effect or immunogenic cancer cell death. As seen before, the PEMBROSARC study [13] aimed to evaluate the combination of metronomic cyclophosphamide, which has demonstrated the ability to suppress Treg lymphocytes and promote the development of T and NK lymphocytes, with pembrolizumab. Another strategy is based on the ability of anthracyclines to generate pro-inflammatory cytokines and damage-associated molecular patterns (DAMPs), activate the production of pro-inflammatory factor IFN1, deplete immunosuppressive cells and boost antigen presentation [32]. Based on this hypothesis, two phase II studies evaluated the combination of doxorubicin with pembrolizumab, in advanced or metastatic sarcomas. The OR rate was 19% in the first study, which included STS and osteosarcomas [33], and 36.7% in the second study, which included only STS [34]. The median PFS was 8.1 months (95% CI, 7.6–10.8) and 5.7 months (95% CI, 4.1–8.9), respectively, to put into perspective the 6.8-month median PFS with doxorubicin alone in a recent study [6]. Importantly, the heterogeneity between those two studies might be explained by the disparity between the two populations (in terms of histological subtypes). Moreover, some of the patients included could have been pre-treated with chemotherapy. In other cancers, immunotherapy alone or in combination with another therapy seems to be more efficient in the early course of the disease possibly due to increasing immune exhaustion over time and treatments. A typical example is small cell lung cancer [35], gastric [36] or triple-negative breast cancer [37], where first-line setting chemo-immunotherapy reached OS improvement whereas it failed in further lines of treatment. The relevance of bringing a chemo-immunotherapy combination to an earlier setting like the first-line treatment of the disease could be questioned. The rationale for the use of cytotoxic chemotherapy in combination with immunotherapy remains substantial and still need to be explored. Some innovative research approaches are ongoing, for example the use of different chemotherapies associated with a single or double blockade doxorubicin + ifosfamide (NCT04356872, NCT04606108), gemcitabine (NCT04577014, NCT03123276, NCT04535713), trabectedin (NCT03138161), eribulin (NCT03899805), etc.

### 2.4. Combination with Radiation Therapy

Radiation therapy is known to synergize with immunotherapy by inducing immune cell death. Indeed, it increases the level of calreticulin, ATP and HMGB1. By inducing these DAMPs, radiation may transform low-immunogenicity STS into so-called “hot” tumors, allowing better ICI efficacy. It is also believed to increase the presentation and diversity of tumor-specific antigens. Radiation-triggered immunogenic cell death releases tumor antigens, tumor cell DNA, cytokines and other danger signals [38]. For example, DNA double-strand breaks activate the pro-inflammatory cGAS–STING pathway [39]. Moreover, genetic damage caused by radiation increases the mutation load and leads to the generation and release of neoantigens, promoting the formation of antigen-presenting cells and thus activating T lymphocytes. The immunological microenvironment altered by irradiation also promotes tumor cell destruction. Changes in the endothelium of tumor blood vessels allow the enhanced migration of the immune cells to the tumor [40,41]. Based on the results of the SARC028 trial reporting encouraging results in UPS and DD-LPS, the SU2C–SARC032 trial aimed to evaluate perioperative pembrolizumab in 105 patients, with these two specific histological subtypes, who had had neoadjuvant radiotherapy [42]. To achieve the best synergies between ICI and radiotherapy, the sequence, dose and fractionation need to be optimized. Preclinical studies have shown that a high-dose per fraction is correlated with a better tumor immune response [43]. In addition to fractionation, one of the most crucial matters is the optimal timing of irradiation. Although there is a debate about sequential or concomitant ICI and radiotherapy, recent clinical data from a PACIFIC trial on non-small cell lung cancer showed survival benefits in the subgroup of patients receiving durvalumab within 14 days after irradiation. This could have been due to the recruitment of newly activated T cells that could destroy both tumors and distant metastases [44]. Although biological springs of the synergy between radiotherapy and immunotherapy still need to be explored, there is a strong rationale for adding ICI to irradiation in sarcoma settings.

Despite the diverse combinations, ICIs are still struggling to find their place in the therapeutic arsenal for the treatment of sarcomas, with variable ORR and PFS depending on the studies (Table 1), which reinforces the need to find effective biomarkers.

## 3. Immune Checkpoint Inhibitors in STS: Challenge of Patient Selection and Stratification

As seen in the previous studies, the greatest challenge remains the improved selection of the patients most likely to benefit from immune therapy. Different hypotheses can be put forward, based on patients’ tumor and/or pharmacological characteristics. Nevertheless, it is crucial to define objective biomarkers, which would allow the selection of the best candidates, as seen in other types of cancers with PDL1 and CPS scores, for example. To date, the investigations are still ongoing, as the biomarkers used to predict responses to ICI in other types of cancers do not seem to be as predictive in sarcomas.

### 3.1. PD1/PD-L1 Expression

PD1/PD-L1 is an efficient biomarker motivating ICI prescription in several cancers, based on results such as PDL1 staining, combined positive score (CPS) or tumor proportion score (TPS) [45]. In the case of STS, some studies suggest that elevated PD1/PD-L1 expression is associated with worse OS, while others suggest it is associated with favorable OS [46,47]. These controversial data might be explained by various confounding factors, such as the use of different antibody clones, different thresholds, the limitations of assays, and the disparity between the histological subtypes included, on top of sampling bias.

### 3.2. Tumor-Infiltrating Lymphocytes (TILs)

A biomarker more recently investigated in immunotherapy is the presence of TILs, which are white blood cells involved in innate and adaptive immunity, located in the tumor or in its stroma. As for PD-L1, the results of investigations remain conflicting, considering it either a positive or negative prognostic factor [48]. The genuine composition and role of these structures remain underexplored and can be different based on the tumor or the patient, either activating or suppressing immunosurveillance. Depending on the techniques used, some of the stromal TILs might be under-evaluated. It has been shown that patients with a greater population of CD8+- or PDL1-positive macrophages are better responders to pembrolizumab, compared to patients with a bigger proportion of immune regulation cells at the baseline. This was confirmed in sarcomas in a correlative analysis of SARC028, which established a correlation between the response to pembrolizumab and high densities of activated T cells (CD8+ CD3+ PD-1+) and an increased percentage of tumor-associated macrophages (TAM) expressing PD-L1 pre-treatment compared with non-responders [49].

### 3.3. Tumor Mutational Burden (TMB)/MicroSatellite Instability (MSI)

Tumor mutational burden (TMB) and microsatellite instability (MSI) are useful biomarkers in other cancer types, as the more mutated the tumor is, the more efficient the immunotherapy is expected to be. Unfortunately, sarcomas have a rather low TMB, with an average of 1.06 mutations/Mb, as reported in TCGA analysis [50]. Nevertheless, this TMB was heterogeneous with some tumors exhibiting a TMB of ≥10 mut/Mb, especially angiosarcomas (7.6%) and UPS (6.7%) [51]. Therefore, some particular histological subtypes may benefit from immunotherapy through a high TMB, such as STS caused by UV-associated mutations: face and scalp angiosarcomas and subcutaneous malignant peripheral nerve sheath tumors (MPSNT) [52]. The Angiosarcoma Project (ASCproject) is a research study that recently reported the results from a cohort of 47 angiosarcomas: 10 cases of the scalp and face had a median TMB of 20.7 mutations per megabase (Mb), and 37 cases of other localizations had a median TMB of 2.8 mutations per Mb [53]. Among the 10 angiosarcomas of the scalp and face, two had received ICI and showed remarkable and durable responses, while the three treated with anti-PD1 from the rest of the cohort did not benefit from it. These results are encouraging despite the very limited sample size. In other types of cancers, the identification of a MSI-high signature status directly allowed the use of pembrolizumab, implying the strength of this predictive factor. The small proportion of MSI in sarcomas (4/1893 sarcomas in the MSK cohort including 1 UPS, 2 uterine leiomyosarcomas and 1 leiomyosarcoma) seems too low to consider the use of this biomarker in routine practices [51].

### 3.4. Tertiary Lymphoid Structure (TLS)

More recent research has consisted of finding immune cell signatures by studying transcriptomic data, allowing us to identify immune cell signature clusters (low, moderated or high), named sarcoma immune classes (SIC). Petitprez and al. have recently demonstrated the correlation of immune cell signatures with responses to anti-PD1 therapy. Hence, patients in SIC-E class, which gathers a high immune activity signature sarcomas are more likely to have an objective response and a better PFS [54]. Some further analyses have demonstrated that class E was characterized by the presence of tertiary lymphoid structures (TLS) that contained T cells and more specifically DC LAMP+ dendritic cells and CD20+ B cells, which are the strongest prognostic factor even in the context of high or low CD8+ T cell and cytotoxic contents. TLS are organized aggregates of immune cells, not found under physiological conditions, but arising in the context of infection, auto immune diseases or, in this case, cancer. Their composition is very similar to that of secondary lymphoid organs, such as lymph nodes (Figure 2). This work discloses the potential of B-cell-rich TLS as a new to with which to select patients. Furthermore, an extended cohort of 48 TLS-positive STS patients out of 240 screened from the PEMBROSARC trial has emerged and resulted in an encouraging outcome. Indeed, among the 35 evaluable patients, the clinical benefit rate (CBR) was 63% (OR = 30%; SD = 33%), in comparison with the 2% OR when analyzing the whole population without prior selection. The median PFS and OS were 4.1 and 14.5 months, respectively [55]. Apart from the promising results of these two studies (Table 2), the CONGRATS study (NCT04095208), still recruiting, includes STS patients with a sarcoma enriched with TLS, evaluating treatment with nivolumab, anti-PDL1, which is associated with anti-LAG3, and relatlimab. The expression of LAG-3 in tumor immune cells has been observed in various tumors. It involves inhibiting a checkpoint on the surface of T cells by blocking lymphocyte activation gene 3. Preliminary data suggest that the dual blockade of LAG-3 and PD-1 has the potential to improve efficacy without substantially increasing toxicity compared to a PD-1 blockade alone. Another study (SPARTO and NCT05210413) evaluates the combination of spartalizumab (anti-PD1) and low-dose pazopanib in solid tumors including TLS-positive STS. Overall, further studies are needed to validate this biomarker, but it seems like another important step in the path of the optimized selection of patients.

## 4. Adoptive Cellular Therapies: A New Opening Door in Sarcomas

As discussed before, one of the principles of immune evasion implies a lack of neoantigens or a defect in the antigen presentation and recognition pathway, preventing patients’ immune systems from generating an adequate immune response to the invasion of the cancer. T lymphocyte activation requires an interaction between a T cell receptor (TCR) and the major histocompatibility complex (MHC). Thus, it is important to educate the host’s immune system by exposing T cells to the tumor antigen for them to develop specific receptors and expand them to develop high-quality and high-avidity antigen-specific T-cell clones. Adoptive cellular therapy is a personalized treatment that involves the isolation of a patient’s own immune cells, their ex vivo modification and expansion and their reinfusion, thus bypassing antigen presentation. Promising results have been attained in hematologic malignancies by using T cells transduced with vectors encoding TCRs recognizing HLA I-restricted antigens or with chimeric antigen receptors (CARs) recognizing cell surface proteins [57,58]. This cutting-edge technology requires investigations to find an antigen that is solely, or predominantly, expressed by tumor cells. The increased difficulty when it comes to sarcomas is due to their high heterogeneity, thus implying that there is a vast diversity of the selected antigens both in terms of histological subtypes but also within a single tumor.

### 4.1. Engineered T Cell Receptor Therapy

To redirect T cells against tumor cells, they can be engineered ex vivo to express cancer-antigen-specific T cell receptors (TCRs), generating products known as TCR-engineered T cells (TCR T). TCRs recognize HLA-presented peptides derived from the proteins of all cellular compartments, and this requires the presence of matched HLA allele subtypes in patients. As seen before, it is crucial to find a reliable neoantigen. Some cancer testis antigens (CTA), such as antigens with MAGE and NY-ESO expression in synovial sarcoma (SS) and myxoid LPS (MLPS), are exceptionally high in quantity and homogenous, ranging from 49% to 82%. While the normal function of the proteins remains elusive, NY-ESO-1 has been shown to interact with MAGE-C1 and may be important in tumor cell proliferation and tumor survival by the inhibition of p53 [59,60]. Afamitresgene autoleucel (afami-cel) and Letetresgene autoleucel (lete-cel) are two experimental therapies based on genetically engineered T cells. They consist of autologous CD4+ and CD8+ T cells that have been genetically modified to express a T-cell receptor (TCR) recognizing MAGE-A4 (afami-cel) or NY-ESO1 (lete-cel) bound to human leukocyte antigen A*02 (HLA-A*02) to induce anti-tumor responses in patients with SS and MLPS expressing those CTA. Afami-cel has been investigated in a phase I and a phase II trial dedicated to SS and MRCL MLPS (SPEARHEAD-1, NCT04044768) that is still recruiting. The preliminary results of the phase II trial was presented at the ASCO meeting in 2021 [61]. Among the 32 patients who received afami-cel at the data cut-off, 25 were evaluable for preliminary efficacy (23 with synovial sarcoma and 2 with MRCLPS). The investigator-assessed responses were for CR (2 patients), PR (8 patients), SD (11 patients) and PD (4 patients). Interestingly, 9 of the 10 responders had ongoing responses at the data cutoff. A pooled analysis from phase I of NCT03132922 and SPEARHEAD-1 gathered the data from 69 MAGE-A4^+^ patients and displayed 36.2% objective responses [62]. The efficacy and safety of lete-cel have also been previously evaluated in a phase I and a phase II clinical (NCT01343043 and NCT02992743). The preliminary efficacy results of phase I [63,64,65] reported impressive responses (50% and 40% objective responses for SS and MLPS, respectively, in patients who had received a high-dose lymphodepletion regimen, including 1 complete response). In addition, responses deepened over time and were durable; responses were ongoing in four patients (two with SS and two with MLPS). These preliminary data demonstrate that afami-cel and lete-cel are efficacious in heavily pre-treated patients (Table 3). Importantly, the safety profile has been favorable, with mainly a low-grade cytokine release syndrome (≤Grade 2) and tolerable/reversible hematologic toxicities being directly correlated to lymphodepletive chemotherapy.

### 4.2. CAR T Cell Therapy

Unlike TCRs, CAR T cells can target any cell surface protein, independently of HLA. One of the main obstacles to the use of this treatment is its limiting toxicity, caused via the release of cytokines caused by the stimulation of the immune system (systemic cytokine release syndrome). Thus, its position remains limited in the therapeutic arsenal of solid tumors, due to the difficulty of identifying a target that does not involve too-severe toxicities in the organs expressing this antigen. Some targets have been tested in sarcomas, including HER2, which is a ligand involved in the Ras/Raf/MEK/ERK1/2 pathway [66]. A phase I/II study studying HER2 CAR-T cell therapy included 19 patients with HER2+ sarcomas and found the OS to be 10.3 months, with an excellent safety profile. A preparation of this therapy with lymphodepleting chemotherapy seems to improve the results [67]. A following study included 10 patients with HER2+ sarcomas (including rhabdomyosarcoma and synovial sarcoma). The patient with rhabdomyosarcoma exhibited a complete response for 6 months [68]. Other targets are being investigated (EGFR (NCT03618381), PDGFRα and GD2 (NCT02107963, NCT04539366, NCT03721068, NCT03635632)) as well as another strategy combining this treatment with immune checkpoint inhibitors (NCT04995003) in order to increase its efficacy.

### 4.3. TIL Therapy

TIL therapy consists of extracting TILs from resected or biopsied human tumors, followed by ex vivo expansion away from the suppressive tumor microenvironment and reinfusion after lymphodepletive chemotherapy. TILs demonstrated the ability to target autologous tumor cells while saving MHC-compatible allogeneic tumor cells or normal autologous cells, in an early experimentation including ten sarcomas [69]. This was confirmed with their use in melanomas, displaying encouraging durable responses, with limited toxicity [70,71,72]. It has been shown that TILs derived from melanoma specifically target multiple antigens, thus standing out as an interesting treatment in the case of heterogeneous tumors, such as sarcomas [73]. Mullinax et al. confirmed that the feasibility of the treatment on sarcomas is of a degree required for clinical use [74]. A few clinical trials including sarcomas are ongoing, including a phase I (NCT04052334) and phase II (NCT03935893) trial. Sarcomas usually having poor lymphocytic infiltration, and some studies aim to assess the safety of combinations; for example, a study involving the use of LTX-315, an oncolytic peptide intended to increase TILs (NCT03725605) [75].

## 5. Conclusions

Immunotherapy with ICI has been reported to cause limited clinical activity in clinical trials. However, these trials included unselected populations of advanced STS patients. Nevertheless, a subset of patients showed remarkable and durable responses, after the failure of many traditional systemic treatments. Furthermore, recent research has restored hope in the possibility of sarcomas being treatable by immunotherapy [76]. One of the major challenges is to define the best candidates for these treatments by a better knowledge of each tumor’s immune component’s microenvironment and by the use of reliable biomarkers and response predictors, in order to optimize personalized medicine plans. The other great challenge is to expand the population that can benefit from them by overcoming immune evasion. Once these two obstacles have been overcome, immunotherapy could become one of the standard treatments for STS.

While the expression of PD-L1 has been reported with variable levels across studies and histotypes, this biomarker has not been found to be associated with the response to immunotherapy. Conversely, the presence of TLS in the primary tumor has been reported to be associated with a higher response rate, longer PFS and survival in both a retrospective study and a prospective study, across histological subtypes. Otherwise, ICI showed high clinical benefit rates in some selected STS histotypes; in particular, ASPS. Given the rarity of each sarcoma subtype, their heterogeneity and the small study sizes, other methodological approaches such as meta-analysis could be relevant. Finally, one of the most exciting approaches in sarcoma immunotherapy is the use of adoptive T cell therapies. By using overexpressed CTA MAGE-A4 and NY-ESO1, genetically modified T cells have been successfully developed to specifically target malignant cells in SS and MRCLS.

Other therapies aim to directly boost immunity by activating pro-inflammatory pathways. One of these involves an agonist of the IL2 pathway, NKTR-214, being combined with nivolumab to treat bone and STS (NCT03282344). Although significant and lasting responses were observed, the lack of a comparative arm is an important bias. Another phase II study is evaluating the addition of IFNy to pembrolizumab. Indeed, preliminary studies have shown that IFNy can activate the expression of MHC class I and therefore the infiltration of T lymphocytes [77]. These approaches seem promising in the activation of the primary immune response and the thwarting of the resistance mechanisms linked to the defect of innate immunity.

In conclusion, even if immunotherapy is not yet applied in the routine treatment of STS (excluding the recent FDA approval of atezolizumab for ASPS), there are promising perspectives based on the better selection of patients (in terms of histotypes and the presence of TLS), innovative therapeutic agents (especially adoptive T-cell therapies) and the combination of immunotherapeutic agents with other therapies.

## Figures and Tables

**Figure 1 cancers-15-01643-f001:**
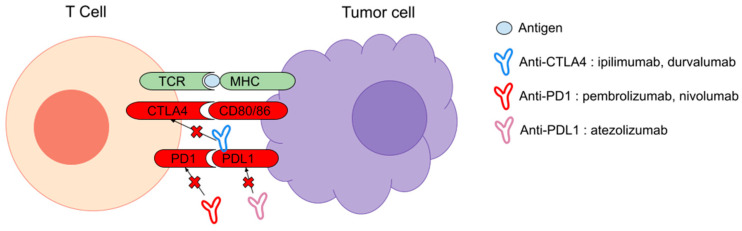
Mechanism of action of the most common ICI.

**Figure 2 cancers-15-01643-f002:**
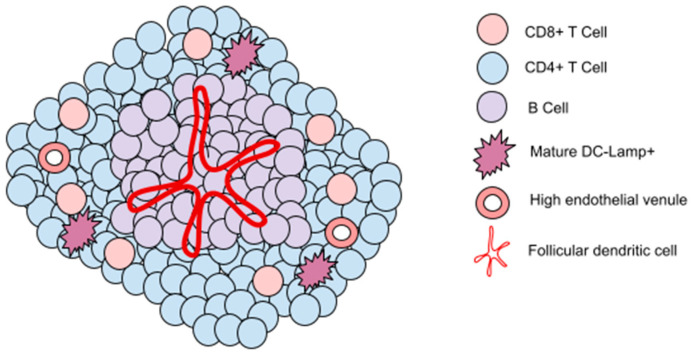
Composition of a tertiary lymphoid structure.

**Table 1 cancers-15-01643-t001:** Responses to checkpoint inhibitors and combinations in sarcomas.

Trial	Target	ICI	Combination Drug	ORR		Median PFS (Months) [95% CI]
PEMBROSARC Toulmonde et al. [12]	57 STS ^1^	Pembrolizumab	Cyclophosphamide	2%		1.4 [1.2–1.4]
SARC028 Tawbi et al. [13]	40 BS ^2^/40 STS ^1^	Pembrolizumab	None	BS ^2^	5%	2 [1.8–2.3]
				STS ^1^	18%	4.5 [2–5.3]
SARC028 (expansion cohorts)	39 DDLPS ^3^ and 40 UPS ^4^	Pembrolizumab	None	DDLPS ^3^	10%	2 [2,3,4]
				UPS ^4^	23%	3 [2,3,4,5]
Alliance A091401 D’Angelo et al. [14]	43 and 42 all sarcoma	Nivolumab +/− ipilimumab	None	Nivo	5%	1.7 [1.4–4.3]
				Nivo + ipi	16%	4.1 [2.6–4.7]
DART Wagner et al. [27]	16 AS ^5^	Nivolumab + ipilimumab	None	25%		NA
Wilky et al. [21]	33 STS ^1^	Pembrolizumab	Axitinib	25%		4.7 [3.0–9.4]
Martin-Broto et al. [29]	58 STS ^1^	Nivolumab	Sunitinib	21%		5.6 [3.0–8.1]
Pollack et al. [31]	37 all sarcoma	Pembrolizumab	Doxorubicin	19%		8.1 [7.6–10.8]
Livingston et al. [32]	30 STS ^1^	Pembrolizumab	Doxorubicin	36.7%		5.7 [4.1–8.9]
Somaiah et al.	57 all sarcoma	Durvalumab + tremelimumab	None	14.3%		4.5 [2.8–6.9]

^1^ STS: soft tissue sarcoma; ^2^ BS: bone sarcoma; ^3^ DDLPS: dedifferenciated liposarcoma; ^4^ UPS: undifferentiated pleomorphic sarcoma; ^5^ AS: angiosarcoma.

**Table 2 cancers-15-01643-t002:** Responses to checkpoint inhibitors in SIC-E and TLS^+^ STS.

Trial	ICI	ORR	
SARC028 [14,54,56]	Pembrolizumab	Overall population (N = 47)	21.2%
		SIC-E Class (N = 10)	50% (with 1 CR)
PEMBROSARC [13,55]	Pembrolizumab	Unselected population (N = 50)	2%
		TLS^+^ STS (N = 35)	30%

**Table 3 cancers-15-01643-t003:** Responses to TCR T cells in MAGE-A4^+^ and NY-ESO ^1^ SS and MLPS: preliminary results.

Drug	Trial	Target	ORR ^7^	Tolerance
Afami-cel	SPEARHEAD-1 NCT04044768	N = 32 28 MAGE-A4^+^ SS + 4 MAGE-A4^+^ MLPS	40% (2 CR ^3^, 8 PR ^4^, 11 SD ^5^, 4 PD ^6^)	59% CRS ^8^ with 95% ≤ Grade 2 and 0% ICANS ^9^
	Pooled analyses from phase 1 NCT03132922 and SPEARHEAD-1 (NCT04044768)	N = 69	36.2%	Not applicable
		59 MAGE-A4^+^ SS ^1^	40.7%	
		10 MAGE-A4^+^ MLPS ^2^	10.0%	
Lete-cel	NCT0134043	N = 45 NY-ESO1^+^ SS ^1^	33% (15/45)	44% CRS ^8^ with 80% ≤ Grade 2
		Cohort 1: high NY-ESO1 expression with cyclophosphamide + fludarabine lymphodepletion	50% (6/12) (1 CR ^3^, 5 PR ^4^, 5 SD ^5^, 1 PD ^6^)	
		Cohort 2: low NY-ESO1 expression with cyclophosphamide + fludarabine lymphodepletion	31% (4/13) (0 CR ^3^, 4 PR ^4^, 7 SD ^5^, 1 PD ^6^)	
		Cohort 3: high NY-ESO1 expression with cyclophosphamide only	20% (1/5) (0 CR ^3^, 1 PR ^4^, 3 SD ^5^, 0 PD ^6^)	
		Cohort 4: low NY-ESO1 expression with cyclophosphamide only	27% (4/15) (0 CR ^3^, 4 PR ^4^, 10 SD ^5^, 1 PD ^6^)	
	NCT02992743	N = 20 NY-ESO1^+^ MLPS ^2^		80% CRS ^8^, with 75% ≤ Grade 2 and 0% ICANS ^9^
		Cohort 1: reduced-dose lymphodepletion	20% (2/10) (2 PR ^4^, 8 SD ^5^, 0 PD ^6^)	
		Cohort 2: standard-dose lymphodepletion	40% (4/10) (4 PR ^4^, 5 SD ^5^, 1 PD ^6^)	

^1^ SS: synovial sarcoma; ^2^ MLPS: myxoid liposarcoma; ^3^ CR: complete response; ^4^ PR: partial response; ^5^ SD: stable disease; ^6^ PD: progressive disease; ^7^ ORR: overall response rate; ^8^ CRS: cytokine release syndrome; ^9^ ICANS: immune effector cell-associated neurotoxicity syndrome.

## Data Availability

Not applicable.

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
