# Peer review of "Immunotherapy for Soft Tissue Sarcomas: Anti-PD1/PDL1 and Beyond"

_cancers, 2023, doi:10.3390/cancers15061643_

Round 1
Reviewer 1 Report
The authors have performed an interesting and in-depth review of the state of the art of immunotherapy for soft tissue sarcomas. However, it would be desirable to better organize the results in structured tables and/or figures to give the reader a visual summary. Furthermore, the introduction of meta-analysis into the work should be considered.
Author Response
Thank you for pointing this out. Therefore, we have organized the results from the use of ICI in TLS positive soft tissue sarcomas and from TCR T-cells ongoing studies in tables (table 2, page 8, subsection 3.4 ; and table 3, page 10, subsection 4.1). We also provided a figure to offer a quick overview of the mechanism of action of ICI (figure 1, page 3, subsection 2.1)
We agree with the importance of suggesting meta-analysis, which we emphasized in the conclusion (line 38-40, page 11, subsection 5).
Reviewer 2 Report
AnyThe manuscript deals with a currently very importatn topic - difficult to trat soft issue sarcoma. It is an up to date presentation of new developments in biological therapy. Well structured, clearly presented. i have no objection to its publication. It is of great practical importance.
Author Response
Thank you for taking the time to review our work and for your comment.
Reviewer 3 Report
The authors have done a thorough and comprehensive examination of the outcomes of clinical trials involving immune checkpoint inhibitors for the treatment of sarcoma. They discuss the current status of immunotherapy agents, either alone or in combination with other therapies, for different subtypes of sarcoma. They emphasize the importance of identifying reliable predictive markers that can help determine the response of individual patients. One question that arises is whether there are any lessons to be learned from the example of ASPS regarding how to determine which patients responded to treatment and which did not.
Overall the review is comprehensive and well-written, making it an excellent resource for journal readers interested in this topic.
Author Response
Thank you for your review. The predictive factors of response within ASPS subtype are indeed an important track that needs exploration. We have, accordingly, added a sentence pointing out the first hypotheses (line 36-38, page 3, subsection 2.1), but also the importance of pursuing the investigations, as there is no clear explanation at this time.
Reviewer 4 Report
Thanks for asking me to review the manuscript.
It is well written, and provides a good overview of Immunotherapy of Soft tissues sarcomas, with appropriate literature review and summary.
It will of interest to the clinicians and researchers managing/researching STS
Author Response

(The authors gave the same response as above.)

Reviewer 5 Report
The review article entitled “Immunotherapy for Soft Tissue Sarcomas: anti-PD1/PDL1 and beyond”, by Fazel et al is focused on the current immunotherapeutic approaches in soft tissue sarcomas. The contents are of interest and the paper is well written.
My major concern is that the structure and the take home message of the review are very similar to a review recently appeared on Biomedicines: Kerrison, W.G.J.; Lee, A.T.J.; Thway, K.; Jones, R.L.; Huang, P.H. Current Status and Future Directions of Immunotherapies in Soft Tissue Sarcomas. Biomedicines 2022, 10, 573. https://doi.org/10.3390/ biomedicines10030573.
The Authors should at least cite this review and emphasize new perspectives.
Minor Comments:
- In the introduction a short sentence illustrating the mechanisms of Immune Checkpoint Inhibitors (ICI) and an appropriate reference could be inserted.
- Line 81-82: have a therapeutically impact.. check english
- ASPS line 85 abbreviation has already been provided in the simple summary.
- Please provide the number of the clinical trial where appropriate
Author Response
The review article entitled “Immunotherapy for Soft Tissue Sarcomas: anti-PD1/PDL1 and beyond”, by Fazel et al is focused on the current immunotherapeutic approaches in soft tissue sarcomas. The contents are of interest and the paper is well written.
My major concern is that the structure and the take home message of the review are very similar to a review recently appeared on Biomedicines: Kerrison, W.G.J.; Lee, A.T.J.; Thway, K.; Jones, R.L.; Huang, P.H. Current Status and Future Directions of Immunotherapies in Soft Tissue Sarcomas. Biomedicines 2022, 10, 573. https://doi.org/10.3390/ biomedicines10030573.
The Authors should at least cite this review and emphasize new perspectives.
Thank you for this suggestion. We have cited this review [76] (line 26, page 11, subsection 5) to incorporate your suggestion. We also added a subsection about TIL therapy (4.3), which emphasizes new perspectives.
Minor Comments:
- In the introduction a short sentence illustrating the mechanisms of Immune Checkpoint Inhibitors (ICI) and an appropriate reference could be inserted.
We have revised the introduction on ICI in subsection 2.1 accordingly and added a simple figure for the readers to better visualize the mechanism of action (figure 1, subsection 2.1).
- Line 81-82: have a therapeutically impact.. check English
We have corrected this error.
- ASPS line 85 abbreviation has already been provided in the simple summary.
We have modified the manuscript to use abbreviations when appropriate.
- Please provide the number of the clinical trial where appropriate
Thank you for pointing this out. We have added the NCTs for the ongoing trials of ICI/chemotherapy combinations (line 13-14, page 5, subsection 2.3), and CAR T cell therapies (line 1-2, page 11, subsection 4.2).
Round 2
Reviewer 1 Report
The two tables and the figure added to the manuscript slightly increased its readability.
However, the authors only mentioned the meta-analysis as a possible further investigation methodology. I would have expected the application of meta-analysis in this work.